# Learning the Partially Dynamic Travelling Salesman Problem

## Abstract

Learning to solve the Travelling Salesman Problem (TSP) using Deep Reinforcement Learning (Deep RL) and Graph Neural Networks (GNNs) has shown promising results for small instances of the problem. We demonstrate that these methods can be extended to solve instances of a partially dynamic variant of the TSP. Solving this partially dynamic variant more effectively exploits the strengths of reinforcement learning and also presents challenges for more established methods of solving the TSP. We show the policies trained using Deep RL outperform modified versions of TSP solvers and heuristics for different distributions of dynamic vertices, including on larger instances than the policies were trained on. This shows the promise of Deep RL for solving this type of dynamic routing problem which is predicted to become of great importance as logistical services become more flexible and responsive to customer demand. Furthermore, our method is a general purpose approach to Deep RL where the problem consists of selecting items from a dynamically-evolving and arbitrarily-sized set.

## 1 Introduction

In the Travelling Salesman Problem (TSP), we are given a list of vertices (representing locations) and distances between them. A tour is the name given to the route which visits every vertex exactly once and returns to the starting vertex. When solving an instance of TSP, the objective is to find the shortest possible tour. It is an NP-Hard combinatorial optimization problem, which means that when solving it there is typically a trade-off between the speed and the exactness of the solution. The most well-known TSP solver, Concorde (Applegate et al., 2006), uses a cutting-plane method Applegate et al. (2001) to solve the problem exactly and there are many heuristic methods (Rosenkrantz et al., 1974; Lin & Kernighan, 1973; Christofides, 2022) which compute approximate solutions in a much shorter time.

There are several papers which have applied Reinforcement Learning methodology to TSP. In Bello et al. (2016), a pointer network (Vinyals et al., 2015) policy is trained using policy gradient methods (Williams, 1992) with a critic baseline (Degris et al., 2012). By sampling the policy network multiple times and selecting the best performing solution, their method produces solutions that are close to optimal for examples of the TSP in 2D Euclidean space with up to 100 vertices. Kool et al. (2019) present an improvement to this method by using an attention-based model and utilising a novel baseline in actor-critic which involves a greedy rollout of the best performing policy. Another improvement is made by Joshi et al. (2022) who use a GNN to encode TSP instances.

All three of these papers make a compelling demonstration of the effectiveness of solving TSP instances with reinforcement learning for small numbers of vertices. They also highlight the immense challenge of trying to improve upon the established methods for solving this problem.

In this paper, we extend the TSP by having a subset of the vertices in the final tour appear dynamically as the tour is in progress. As we are looking to extend models used for reinforcement learning on the static TSP, we devise a partially dynamic extension to the TSP (PDTSP) which keeps the problem as pure as possible. In this formulation, the only additional constraint added is that dynamic vertices cannot appear in the tour before their assigned arrival time. The objective remains the same as the TSP, finding the shortest possible tour.

The challenge associated with the partially dynamic TSP is that routing decisions need to be made while the route is currently in progress. This makes it important that the solution method is capable of producing updated solutions quickly when new vertices appear in the problem. An optimal solution to this problem, i.e. the shortest tour visiting all of the static and dynamic vertices, is highly unlikely to ever be achieved by any solution method because it would require not only routing optimally through the vertices which are currently known but also routing so that dynamic vertices can be incorporated optimally into the route.

The key benefit of RL in this situation is that it can learn a solution that is adapted to the underlying distribution of dynamic vertices in the problem it is being applied to. This is something that existing TSP solvers cannot be expected to achieve. They have been specifically designed so that they perform well on TSP problems in which the vertices are uniformly distributed in the space.

The RL method works by training a neural network policy using gradient descent on simulated examples of the PDTSP to optimise tour length. These simulated examples enable the RL method to learn tours which outperform modified versions of existing solution methods because the RL method not only learns how to create efficient tours in the TSP but also how to best anticipate the arrival of new vertices into the problem.

From a practical point of view, considering a dynamic variant of the TSP is useful because as delivery and logistic operations continue to develop, they are being expected to operate on shorter timescales (Hildebrandt et al., 2023). Some concrete examples are same-day delivery services, online food-delivery, ride-hailing apps, emergency repair services and emergency medical services. Being able to make alterations to routes quickly in response to new locations appearing in a schedule is clearly desirable in these applications.

In the literature, there are existing partially dynamic extensions to the travelling salesman problem. For example, the Partially Dynamic Travelling Repairman Problem (Larsen et al., 2002), the dial-a-ride problem (Cordeau & Laporte, 2007) and the Dynamic Vehicle Routing Problem with Stochastic Requests (DVRPSR) (Zhang et al., 2023). Each of these problems introduce additional constraints such as time windows, time horizons, vehicle capacities, multiple vehicles and different types of vertices (i.e. pick-up vertices and drop-off vertices). The objectives of these extensions typically differ from the pure TSP objective of finding the shortest tour, instead optimising the waiting times of customers or the total number of customers served. For this initial extension of the TSP, we choose to demonstrate efficacy on two simple cases of the PDTSP, one with a unimodal spatial distribution and one with a bimodal spatial distribution where the probability shifts from one mode to another as the tour is built.

Our contributions are (a) a new partially dynamic travelling salesman environment, (b) a demonstration that the reinforcement learning methods and models developed for the static TSP outperform static solvers and heuristics which we modify to work in the PDTSP, and (c) a demonstration of effective performance on larger instances than those in the training sets. This is particularly important because it means that policies trained on small instances due to compute concerns can be utilised effectively on larger real-world instances.

## 2 PARTIALLY DYNAMIC TSP ENVIRONMENT

The problem we consider is a dynamic variant of the Travelling Salesperson Problem (TSP) in which a subset of the vertices in the problem are static and their locations are known prior to commencing the tour. The remaining vertices are dynamic and their locations are revealed as the tour progresses. We call this variation a Partially Dynamic Travelling Salesperson Problem (PDTSP).

Just like in the static TSP, the objective is to minimize the total length of this tour, $\Pi$, given by:

$$L(\Pi|s) = ||x_{\Pi_n} - x_{\Pi_1}||_2 + \sum_{i=1}^{N-1} ||x_{\Pi_i} - x_{\Pi_{i+1}}||_2, \tag{1}$$

where $x_i \in [0,1]^2$ are the vertex co-ordinates. Here $s$ is used to denote a particular instance of the PDTSP, $\Pi$ is used to denote a tour which is a permutation of the vertices, and $N$ is the total number of static and dynamic vertices.

For a particular instance of PDTSP, there are a set number of static and dynamic vertices. The static vertex co-ordinates are sampled uniformly in the unit square. The dynamic vertex co-ordinates are sampled according to a seperate spatial distribution over the unit square. When simulating data for this paper, we consider unimodal and bimodal distributions. The arrival times, $t_a \in \mathcal{T}_a$ of the dynamic vertices are sampled without replacement from a categorical distribution with support $[2, N-1]$.

Time in the PDTSP is discrete and each time step corresponds to visiting a new vertex on to the tour. At time $t = 1$, the salesperson will be located at the starting vertex of its tour. The agent will observe the state of the problem and select the next vertex in its tour using the policy network. At time $t = 2$, the salesperson will be located at the vertex selected at $t = 1$ and so on until the salesperson arrives back at the starting vertex and the tour is complete. If the current time corresponds to an arrival time of one of the dynamic vertices, $\exists t_a \in \mathcal{T}_a : t_a = t$, then that vertex will arrive in the problem at time $t$ and can be selected by the agent at time $t$ or any subsequent time step. Crucially, the agent cannot alter steps of the tour which have been completed before the current time.

In the static TSP problem, the choice of starting vertex is arbitrary as the final tour remains the same regardless of which vertex it begins from. In the PDTSP, the order in which vertices are visited in the final tour matters because the dynamic vertices only appear in the problem after a certain number of vertices have been visited. For this reason, when generating instances of PDTSP, the starting vertex of the tour is fixed.

## 3 MODELLING THE PROBLEM

### 3.1 REPRESENTING THE PROBLEM AS A DYNAMIC GRAPH

We represent the PDTSP as a dynamic graph $\mathcal{G} = \{G_{t_a} : t_a \in \mathcal{T}_a \cup 1\}$. Each graph $G_{t_a} = (V_{t_a}, E_{t_a}) \in \mathcal{G}$ is a complete graph and is referred to as a snapshot of the dynamic graph. The vertices $v_i \in V_{t_a}$ have co-ordinates $x_i$ and the edges $e_{ij}$ have weights corresponding to the Euclidean distance between the two connected vertices, $||x_i - x_j||_2$.

The first snapshot, $G_1 = (V_1, E_1)$, is the initial graph consisting of the static vertices which are present at the start of the problem. The next snapshot, $G_{t_{a_1}} = (V_{t_{a_1}}, E_{t_{a_1}})$ occurs when the first dynamic vertex arrives in the problem, $V_{t_{a_1}} = V_1 \cup \{v_{n+1}\}$, the edge set is extended to include edges from all other vertices to the new dynamic vertex. The snapshots follow this pattern until all dynamic vertices have been added to the graph.

Before being input to the neural network model, a graph sparsification heuristic is applied to each graph snapshot $G_{t_a}$. The heuristic works by finding the set of $k$-nearest neighbours of each vertex in the graph and removing edges to all the other vertices in the graph which do not belong to this set. Joshi et al. (2022) show that this improves the performance of the neural network model.

### 3.2 NEURAL NETWORK ARCHITECTURE

The policy is embodied by a neural network with an Encoder-Decoder architecture (Hamilton et al., 2017) as depicted in Fig. 1. The encoder component takes as input the current state of the PDTSP problem represented as a graph and outputs a $d$-dimensional embedding of each vertex in the graph. The decoder component takes this embedding as input and outputs a probability distribution over actions which correspond to unvisited vertices in the PDTSP.

#### 3.2.1 ENCODER

The encoder receives as input the vector of vertex co-ordinates, $x_i \, \forall \, v_i \in V_{t_a}$, the vector of edge weights, $||x_i - x_j||_2 \, \forall \, e_{ij} \in E_{t_a}$, and an adjacency matrix of the graph $G_{t_a}$.

As in Joshi et al. (2022), the encoder is a GNN. The vertex and edge features are first projected into $\mathbb{R}^d$ by linear layers to obtain feature vectors $h_i^0$ and $e_{ij}^0$. This keeps the dimension consistent in the message passing component. In the message passing component, the vertex and edge features are updated by a stack of $L$ message passing layers. Each message passing layer updates vertex

features by a weighted aggregation of its own features with those of its neighbours in the graph. The functional form of the message passing layers for vertex features is given by,

$$h_i^{\ell+1} = h_i^\ell + ReLU\left(\text{Norm}\left(U^\ell h_i^\ell + \text{Aggr}_{j \in \mathcal{N}_i}\left(\sigma(e_{ij}^\ell) \odot V^\ell h_j^\ell\right)\right)\right), \tag{2}$$

Here, $h_i^\ell$ is the vector of vertex features of vertex $v_i$ output by message-passing layer $\ell$. Similarly $e_{ij}^\ell$ is the vector of edge features of the edge between vertices $i$ and $j$ output by message-passing layer $\ell$. $U^\ell$ and $V^\ell$ are network parameters, $\sigma$ indicates the application of a sigmoid function, $\text{Aggr}$ indicates the application of an aggregation function over the set of neighbouring vertices $\mathcal{N}_i$, $ReLU$ is the rectified linear unit function and $\text{Norm}$ indicates the application of a batch normalization layer. The $\odot$ symbol is the element-wise multiplication operator.

The edge features are updated by a weighted aggregation of the edge features with the features of the two vertices connected by the edge. The functional form is given by,

$$e_{ij}^{\ell+1} = e_{ij}^\ell + ReLU\left(\text{Norm}(A^\ell e_{ij}^\ell + B^\ell h_i^\ell + C^\ell h_j^\ell\right), \tag{3}$$

$e_{ij}^\ell$ and $h_i^\ell$ are as above, as are the $ReLU$ and $\text{Norm}$ functions. $A^\ell, B^\ell$ and $C^\ell$ are network parameters. For a network with $L$ message passing layers, the final output of the encoder, $h_i^L$, is a $d$-dimensional vertex embedding for each vertex in the PDTSP graph.

When a new arrival occurs at time $t_a$, we obtain a new graph $G_{t_a}$, with a new set of vertices and edges. This graph is input to the encoder and a new embedding is obtained to feed to the decoder in the subsequent time steps until another arrival occurs.

### 3.2.2 DECODER

The decoder component of the network comes from Kool et al. (2019). It operates sequentially, producing a probability over vertices at each timestep of the tour, $t \in \{1, ..., N_{Total}\}$.

At each timestep $t$, the input to the decoder are the vertex embeddings output by the encoder $h_i^L$. The final edge embeddings from the encoder are not used directly by the decoder. The graph embedding, $\bar{h}_i^L$, is obtained by aggregating the vertex embeddings. This is concatenated with the vertex embeddings of the previous vertex in the tour $h_{\Pi_{t-1}}^L$ and the first (and hence last) vertex in the tour $h_{\Pi_1}^L$ to create a context vector, $h_c$,

$$h_c = [\bar{h}^L, h_{\Pi_{t-1}}^L, h_{\Pi_1}^L]. \tag{4}$$

This context vector is then used as the sole query vector in a multi-headed attention (MHA) layer with $M$ heads. The value and key vectors are the vertex embeddings $h_i^L$,

$$q_c = W^Q h_c, \; k_i = W^K h_i^L, \; v_i = W^V h_i^L, \tag{5}$$

where $W^Q$, $W^K$ and $W^V$ are network parameters. The output of this attention layer is an updated context vector $h_{c'}$,

$$h_{c'} = \sum_{m=1}^M W_m^O h_{im}, \tag{6}$$

$$h_{im} = \sum_i \text{Softmax}\left(\frac{q_c^T k_i}{\sqrt{d_k}}\right) v_i, \tag{7}$$

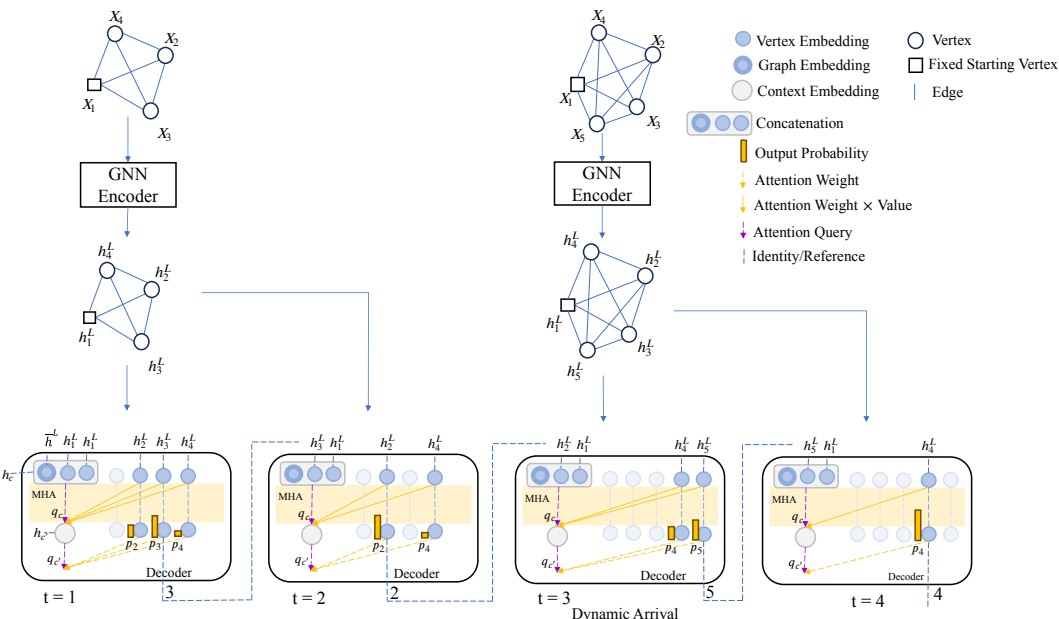

Figure 1: Diagram showing how the neural network model works for a simplified instance of PDTSP in which there are 4 static vertices and 1 dynamic vertex. For this instance, the model outputs a tour $\Pi = (1, 3, 2, 5, 4)$. The initial vertex embeddings from the GNN are input to the Decoder, the embedding of the fixed first vertex is input along with a graph embedding to a context vector. MHA is used to update the context vector with the vertex embeddings, then another round of attention is used to obtain probabilities of the next vertex in the tour. At time step $t = 3$, the dynamic vertex arrives and a new embedding is created using the encoder.

Here, $W_m^O$ are network parameters applied to the output of each of the $M$ attention heads. $h_{im}$ is the output of the $m^{th}$ attention head which is calculated by calculating the attention weights using the query and key vectors, taking a softmax, multiplying by the value vector and summing the results. The updated context vector is input to another MHA layer, again with $M$ heads, but this time we are only interested in the attention weights. The attention weight of each vertex is taken and clipped to lie within the range $[-C, C]$ to obtain log-probabilities of each vertex,

$$u_i = \begin{cases} C \cdot tanh\left(\frac{q_{c'}^T k_i}{\sqrt{d_k}}\right) & \text{if} \quad i \neq \Pi_{t'} \quad \forall t' < t \\ -\infty & \text{otherwise.} \end{cases} \tag{8}$$

Here $d_k = \frac{d_{h_{c'}}}{M}$ is the query/key dimensionality and $q_{c'}$ is the query vector associated with the updated context. Vertices which are already part of the tour are masked. These logits are then input to a softmax function to obtain the probability of selecting each vertex ($\theta$ represents the network parameters),

$$p_\theta(\pi_t | \pi_{1:t-1}) = \frac{e^{u_i}}{\sum_j e^{u_i}}. \tag{9}$$

## 4 TRAINING THE MODEL

Each environment in a training batch shares the same number of static and dynamic vertices in its PDTSP. When each batch is generated, the number of static and dynamic vertices for every environment in that batch is randomly sampled from a range which is set prior to training. This method of generating training batches means that each individual batch is easier to work with but

the policy still experiences a range of problem instances across batches in terms of the total number of vertices and the proportion of vertices which are dynamic.

Each environment in a batch starts at time step $t = 1$. The state corresponding to the current feature graph of each environment is input to the encoder-decoder network and a probability distribution over vertices is output for each environment. A vertex is sampled for each environment and then the state of each environment is updated to add this vertex to the tour and the time step of that environment is incremented by 1. The set of arrival times, $\mathcal{T}_a$, for each environment is then checked to see if $\exists t_a \in \mathcal{T}_a : t_a = t$. The environments which have an arrival are paused. The state of a paused environment is fixed until an arrival has occurred for each environment in the batch. At that point, the graphs for the batch are updated with their new vertex arrival and the batch is input to the encoder and a new graph embedding is produced for each environment. The environments are unpaused and then new vertices continue to be selected until the next arrival happens. The reason for organizing simulations this way is to take advantage of GPU hardware which is typically much faster when calculating a batch of inputs rather than inputs one at a time. [1]

The model is trained by a policy gradient algorithm in which the loss is given by:

$$\mathcal{L}(\theta|s) = \mathbb{E}_{p_\theta(\Pi|s)}[L(\Pi)], \tag{10}$$

the expectation of the tour length $L(\Pi)$, where $p_\theta(\Pi|s)$ is the probability distribution obtained from the policy. The policy gradient is then given by:

$$\nabla\mathcal{L}(\theta|s) = \mathbb{E}_{p_\theta(\Pi|s)}[(L(\Pi) - b(s))\nabla \log p_\theta(\Pi|s)]. \tag{11}$$

These expectations are approximated by the sample mean over the batch of training data. The baseline $b(s)$ is obtained by running a previous version of the policy in which actions are selected greedily rather than by sampling. The previous version of the policy is compared with the current version at the end of each epoch and if the difference between rewards obtained by the current policy is a statistically sufficient improvement on the previous version as determined by a $t$-test, then the baseline model is replaced by the current policy and training resumes.

## 5 EXPERIMENTAL SETUP

In this paper, we look at two training sets. In the first, the spatial distribution of dynamic vertices is a unimodal bivariate Gaussian distribution. From now on, the policy trained on this data will be referred to as the Unimodal policy. In the second the spatial distribution of dynamic vertices is a bimodal mixture of Gaussian distributions and will be referred to as the Bimodal policy.

With bimodal arrivals, the probability of sampling from one mode of the distribution has higher probability during the first half of the tour and the other mode has higher probability during the second half of the tour. This is designed to mimic an example situation that could arise in a practical application in which demand could shift from a commercial location to a residential location as time progresses.

The unimodal distribution has mode $(0.2, 0.2)$ and covariance matrix $0.1 \times I$, where $I$ is the identity matrix. The probability density is illustrated in Fig. 2a. The bimodal distribution is a mixture of two Gaussian distributions with modes at $(0.2, 0.2)$ and $(0.8, 0.8)$ and both having covariance matrices $0.1 \times I$. In the first half of the tour the probability of sampling from the Gaussian distribution at $(0.2, 0.2)$ is double that of the other and vice versa for the second half of the tour. The probability density is illustrated in Fig. 2b.

For both of these training sets, the number of static and dynamic vertices for each batch are both sampled in the range $[20, 50]$. This means that the smallest instance during training will be a 40

---

[1]This simulation process does not affect the training of the network using gradient descent. For a batch of simulations, the final output for each element of the batch is a list of vertex selections and the corresponding probability distribution from which the selections were drawn. The time steps for which the environment sat paused are simply removed for each of the different environments.

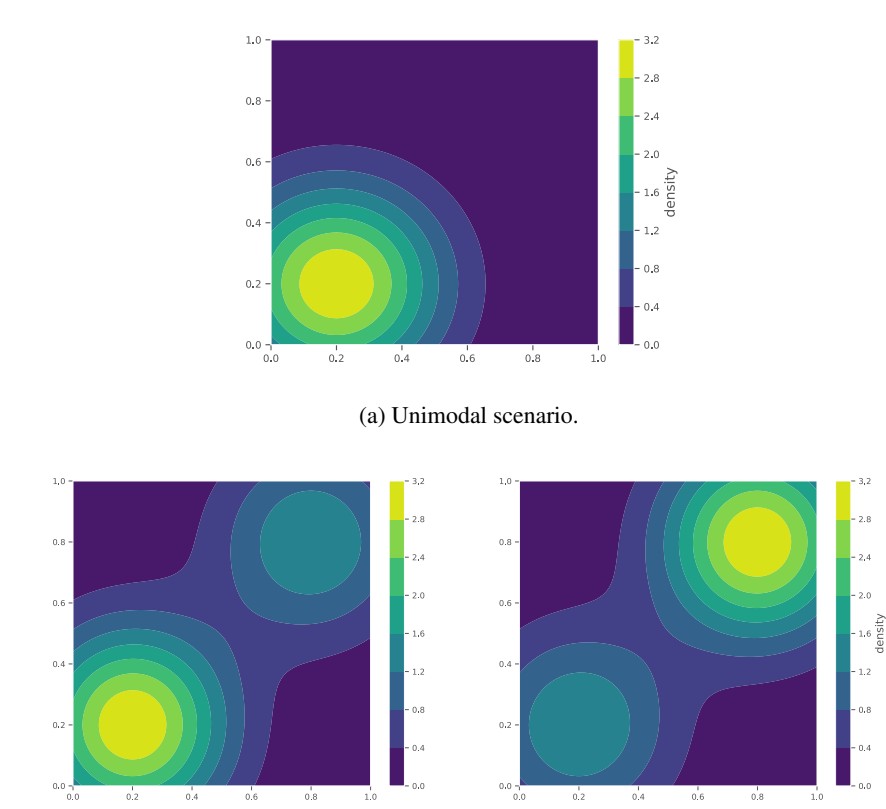

(a) Unimodal scenario.

(b) Bimodal scenario showing the distibution for the first half of the tour on the left and the second half of the tour on the right.

Figure 2: Spatial distributions for the dynamic vertices in the two PDTSP scenarios.

vertex instance and the largest 100. The number of static and dynamic vertices are sampled separately so batches in training will have a variety of different proportions of dynamic vertices. For each training set, the model is trained on a total of $1,600,000$ training instances. These instances are split between 100 epochs of training and each epoch is split into batches of size 16.

In these training runs, the model is built with the following hyperparameters. The encoder consists of 3 message passing layers with embedding dimension, $d = 128$. Max functions are used for aggregation in the message passing layers and also for the graph embedding. The probabilities output by the decoder are clipped with parameter $C = 10$. The total number of model parameters for this set of hyperparameters is $364,544$.

The trained models are evaluated on four test sets. Two each for the unimodal and bimodal model. The test sets for each of the models shares the respective spatial distributions. For each model, there is a test set consisting of sets of $1,280$ PDTSP instances with $40, 80, 120, 160, 200$ and $500$ (for 500 vertices, 128 instances are used) total vertices with a $50/50$ split between static and dynamic vertices. The other test set for each model consists of sets of $1,280$ PDTSP with 100 total vertices where each set has a different proportion $\{0, 0.2, 0.4, 0.6, 0.8\}$ of vertices which are dynamic.

As an optimal solver can not exist for this problem without being given knowledge of future arrivals, the models and comparison methods are evaluated based on average tour length. At evaluation, the policies are queried greedily to better demonstrate the efficacy of the learned policies.

Table 1: Running times for the different methods on different sizes of PDTSP (format static vertices:dynamic vertices) The times given are the time in seconds taken to run 40 instances. The Rerun Concorde, Rerun Insertion and Model CPU methods were ran on a Intel(R) Xeon(R) Gold 6248R CPU @ 3.00GHz processor. The Model GPU method was ran on a NVidia L40 GPU.

| Instance | Rerun Concorde | Rerun Insertion | Model CPU | Model GPU |
|----------|----------------|-----------------|-----------|-----------|
| 20:20    | 3.36           | 0.35            | 11.54     | 0.65      |
| 40:40    | 25.33          | 1.09            | 37.07     | 1.06      |
| 60:60    | 70.43          | 2.40            | 98.20     | 2.81      |
| 80:80    | 176.72         | 4.30            | 175.66    | 4.97      |
| 100:100  | 348.90         | 6.98            | 303.2     | 9.01      |

## 5.1 COMPARISON METHODS

It is necessary to adapt some of the heuristic methods for the Static TSP so that they can be used to solve the partially dynamic TSP. The first method we call Concorde plus insertion. In this method, the Concorde solver is used to compute an optimal tour on the static vertices in the problem. When a dynamic vertex arrives, it is inserted into the tour in the lowest cost position between any two vertices in the tour which have not yet been visited. The cost of insertion of dynamic vertex $k$ between any two vertices $i$ and $j$ is calculated as,

$$c_{ijk} = d_{ik} + d_{jk} - d_{ij}. \tag{12}$$

where $c$ is the cost and $d$ is the distance between the two vertices in the subscript.

The other two methods allow for more adaptation of the tour in response to an arrival. These methods we call rerun insertion and rerun Concorde. These methods involve using either a nearest insertion heuristic or Concorde solver to re-plan the tour through the vertices which have not yet been visited each time a new arrival happens. These two methods give more flexibility as the whole remaining tour can be altered in response to an arrival and not just the position of the new dynamic vertex.

## 6 EXPERIMENTAL RESULTS

For the unimodal policy, Figure. 3a shows the performance of the policy on the test set with different proportions of dynamic vertices. The policy is outperformed by the competing methods for the instance with 0 dynamic vertices but this is expected as this is the static variant of the TSP. When the proportion of dynamic vertices is increased to 0.2 then the policy starts to slightly outperform the competing methods with the gap increasing as the proportion of dynamic vertices increases.

Figure. 3b shows the performance of the policy on the test set of problem instances with different total numbers of vertices. The policy outperforms each of the competing methods for the different numbers of vertices tested. Encouragingly for the applicability of these models, the trend continues for instances with more vertices than was seen in the training set.

There is a similar story when testing the bimodal model, the corresponding plots are shown in Figures. 3c and 3d. The only difference being that the tours tend to be slightly longer on average and the bimodal model is outperformed by the method which reruns Concorde after each new arrival for problem instances with a 0.2 proportion of dynamic vertices. For larger proportions, the model comes out on top with the gap in tour length increasing with the proportion of dynamic vertices.

The performance of the policies evaluated on a mismatched training set (Policy (Bimodal data) in Figures. 3b and 3a and Policy (Unimodal data) in Figure. 3d and 3c) outperforms the comparison methods in both cases. This suggests that the policy is not just learning where dynamic vertices are likely to arrive but also how to build tours which are better at anticipating new arrivals wherever they occur. There is still a small performance gap favouring the policy with matching training data demonstrating that having some prior knowledge of the true distribution of dynamic vertices does benefit the policy.

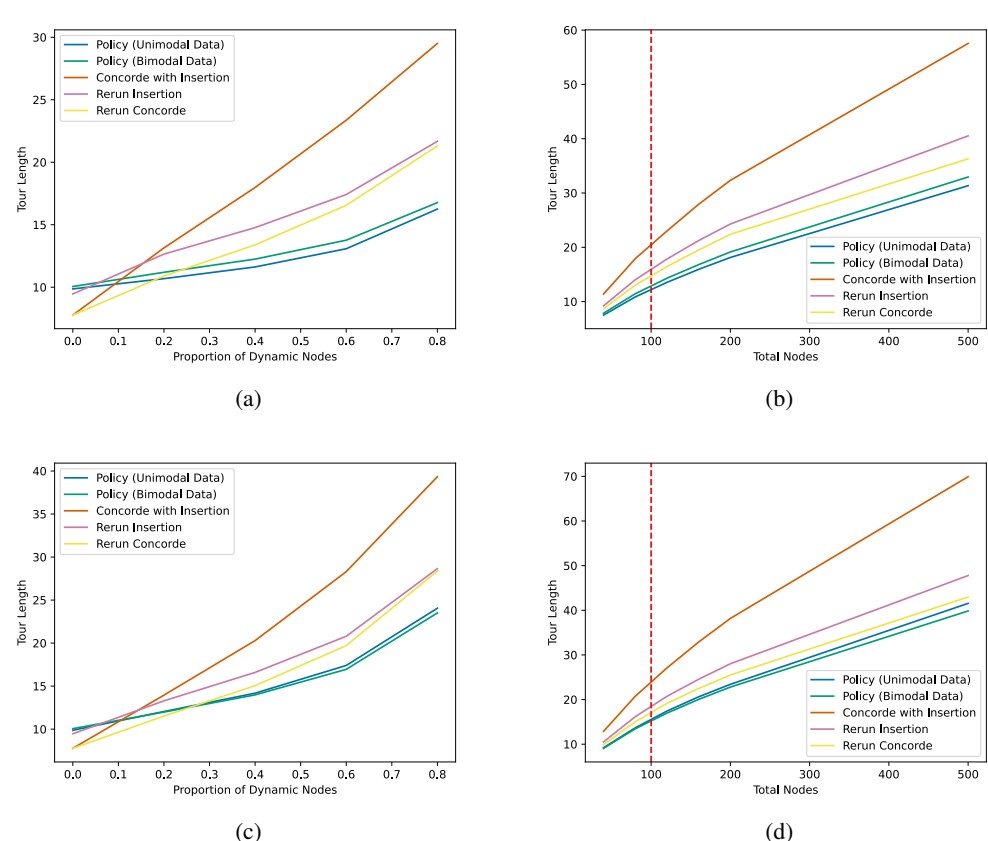

(a)                                    (b)

(c)                                    (d)

Figure 3: Average tour length of the policy evaluated on unimodal (a,b) and bimodal (c,d) test sets compared with comparison methods, including the policy trained on the other data distribution. The red dashed line in plots b and d shows the size of the largest instance in the training set for the RL policies.

The running times of the different methods are shown in Table. 1. Running on a CPU, querying the RL model takes a similar amount of time to complete to rerunning concorde with rerunning the insertion heuristic being much faster. When the RL model is queried on a GPU it is much faster and comparable in speed to rerunning insertion.

# 7 DISCUSSION AND CONCLUSION

The results show that for the PDTSP, the trained policy is able to outperform comparison methods including one which re-solves for an optimal tour each time a new vertex is added. This suggests that the trained policy is not only learning what features make a good tour in TSP but is also learning to create tours which are anticipatory in the face of dynamic vertices. The evaluation time of the RL model is very fast on specialised hardware suggesting that it would be able to make live routing decisions.

The experiments do not look at the performance of the unimodal policy on the bimodal test sets or vice versa. This is because generalizing between spatial distributions that have changed in such a dramatic fashion is not something that is expected in practice. The bimodal scenario already shows that the model is equipped to deal with more realistic changes in the spatial distribution reflecting what might happen in a city when people move from work to home for example. In light of this, we believe that the best direction of future research would be to consider the robustness of a trained policy to small fluctuations in the spatial or temporal distributions of dynamic vertices. Also of

interest would be the possibility of training a baseline policy which is capable of adapting quickly to new spatial distributions corresponding to different cities for example.

Another encouraging aspect of these results is that the trained policies perform well on instances with more vertices than the largest instances in the training sets. This is important because it suggests that the trained policies can be utilised on large instances of the dynamic TSP without requiring prohibitively expensive training on larger instances.

## 8 REPRODUCIBILITY STATEMENT

To enable reproduction of the methods and results in this paper, we attach the code base used during this project. The code base includes test set .txt files which contain the data used to produce the results.

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
