# OpenReview forum: "Learning the Partially Dynamic Travelling Salesman Problem"
_ICLR.cc/2025/Conference — Submitted to ICLR 2025_

### Official Review · Reviewer_5xbT · 2024-10-17

**Soundness:** 3
**Presentation:** 1
**Contribution:** 2
**Rating:** 3
**Confidence:** 4

**Summary:**

This paper presents an approach to solving the Partially Dynamic Travelling Salesman Problem (PDTSP) using Deep Reinforcement Learning (Deep RL) and Graph Neural Networks (GNNs). The authors extend the traditional TSP by introducing a dynamic component where a subset of vertices appears during the tour's progress. They train a GNN-based encoder-decoder architecture on simulated instances of PDTSP and demonstrate that the learned policy outperforms compared methods, especially as the proportion of dynamic vertices increases.

**Strengths:**

The focus on extending the TSP by introducing partially dynamic vertices is relatively less explored in the field of ML4CO (especially ML4TSP), which provides a fresh and interesting direction for the well-studied problem. Also, as preliminary work to combine and apply existing RL models to such a new problem setting is generally appreciated.

**Weaknesses:**

1. While concentrating on the fresh variant of TSP (i.e., PDTSP) is good, the technical contribution proposed in this paper seems somewhat limited. The used encoder-decoder architecture with GNNs and RL training scheme for routing problems are largely dependent on existing methods. The modifications for adaptation to PDTSP are intuitive, that is, by updating the graph instance input once a dynamic vertex appears and generating a new embedding for it. Perhaps more task-specific (even minor) designs of the model or searching tools for better adaptation are desired.
2. The attached repository heavily inherits from [1]. Incremental development to existing works is welcomed; however, I suggest the authors make the demo repo clearer by, for example, updating (at least) a new README that explicitly introduces your work/code or providing compact instructions on how to reproduce your results, instead of leaving the confusing files that adhere to the approach you base your methods on. Note that this is optional.
3. The model is trained on PDTSP with a scale range of $N \in [40, 100]$ and is evaluated at sizes up to 500. Though the generalization performance seems promising, the model's scalability is still questionable because most RL-based neural solvers hardly scale to TSP with more than 100 nodes [2,3,4]. Thus, if the model has difficulty learning on $N > 100$, a natural suspicion arises about the effectiveness of the proposed policy on $N > 500$.
4. As stated by the authors, the main contribution of this paper includes "a demonstration that the reinforcement learning methods and models developed for the static TSP outperform static solvers and heuristics which we modify to work in the PDTSP." To my understanding of this claim, the authors expect to verify the applicability and effectiveness of a mainstream learning pipeline (GNN+RL) to tackle the new problem setting (PDTSP) with empirical results. In this context, it is reasonable to intuitively utilize existing methods as step-wise local solvers in the dynamic progress. Along this path, the experiments are incomplete (only proposed methods and Concorde/insertion). The authors should incorporate more successful neural TSP solvers (optionally for your reference [5-10]) into the PDTSP environment for comparison (maybe in the same way of your rerun-X, where X can be LKH, SL+GNN models, generative models, etc.), in support of your preferred RL approach as a local predictor over other methods.
5. The calculation of the current metric for evaluation is a bit obscure and limited. Although there is no optimal solver for the specific PDTSP task yet, a global (near-)optimal tour as a reference is still an important perspective, as conventionally included in the neural solution of TSP literature. For example, perform Concorde or Gurobi or LKH, etc., on the test instances with all dynamic vertices revealed and fixed, and calculate the performance gap between the reference and your methods. It's not a problem if there is a considerable gap, but this makes your evaluation more complete.
7. The presentation of experimental results is somewhat rough and not informative enough. By convention [2-10], the authors are encouraged to establish several ***main*** test sets (e.g., Unimodal/Bimodal-small/large, etc.) and report accurate results (instead of figures for trends only) of compared methods in a table containing tour length, optimality gap, and solving time to make the results more concrete. The current experiments are more likely ablation studies on node number and proportion of dynamic vertices.

**References:**

[1] Learning the travelling salesperson problem requires rethinking generalization.

[2] Attention, Learn to Solve Routing Problems!

[3] POMO: Policy Optimization with Multiple Optima for Reinforcement Learning.

[4] Sym-NCO: Leveraging Symmetricity for Neural Combinatorial Optimization.

[5] DIMES: A Differentiable Meta Solver for Combinatorial Optimization Problems.

[6] An Efficient Graph Convolutional Network Technique for the Travelling Salesman Problem.

[7] T2T: From Distribution Learning in Training to Gradient Search in Testing for Combinatorial Optimization.

[8] BQ-NCO: Bisimulation Quotienting for Efficient Neural Combinatorial Optimization.

[9] Unsupervised Learning for Solving the Travelling Salesman Problem.

[10] Graph Neural Network Guided Local Search for the Travelling Salesperson Problem.

**Questions:**

Please see the weaknesses section.

---

### Official Review · Reviewer_o8dq · 2024-10-19

**Soundness:** 2
**Presentation:** 1
**Contribution:** 2
**Rating:** 3
**Confidence:** 4

**Summary:**

The authors introduce a partially dynamic TSP variant, where some vertices appear dynamically as the tour progresses, and propose a DRL method to solve it. The experimental results indicate the proposed method outperforms heuristics and solvers adapted to this TSP variant.

**Strengths:**

- Extending neural combinatorial optimization methods to dynamic problem settings is an underexplored research topic.
- Targeting a challenging problem setting since "An optimal solution to this problem, i.e. the shortest tour visiting all of the static and dynamic vertices, is highly unlikely to ever be achieved by any solution method".
- RL suits this problem since "it can learn a solution that is adapted to the underlying distribution of dynamic vertices in the problem it is being applied to".
- Promising experimental results.

**Weaknesses:**

- The authors claim that the literature contains partially dynamic extensions to TSP. However, they introduce a novel and simpler variant with unclear motivation.

- The authors adopt synthetic datasets with unimodal and bimodal spatial distribution. However, the practical relevance of such distributions is unverified. It is preferable to evaluate the method using real-world instances or at least synthetic datasets with practical relevance.

- The proposed method simply adapts prior methods to dynamic settings and lacks solid technical contribution.

- This work targets a single COP and lacks discussions of generality.

- Lacking references to the recent related works.

- Writing should be improved.

**Questions:**

Please see the weaknesses.

---

### Official Review · Reviewer_XAdf · 2024-11-03

**Soundness:** 2
**Presentation:** 2
**Contribution:** 2
**Rating:** 3
**Confidence:** 5

**Summary:**

This paper proposes a partially dynamic travelling salesman problem (PDTSP) and apply a deep reinforcement learning (DRL) and graph neural network to solve it. The key idea is to devise a partial dynamic extension to the TSP and represent the PDTSP as a dynamic graph. Experiments show that the effectiveness and generalization of the DRL approach on the PDTSP.

**Strengths:**

1. This paper proposes a partially dynamic travelling salesman problem, which is an extension of TSP.
2. A demonstration that the reinforcement learning methods and models designed for the static TSP outperform modified static solvers and heuristics for the PDTSP.

**Weaknesses:**

1. The writing and organization of this paper could be enhanced for better clarity.
2. The focus on the partially dynamic version being solely on TSP may limit the demonstration of deep reinforcement learning (DRL) capabilities, which could also be applied to the closely related capacitated vehicle routing problem.
3. The selection of baselines is limited, incorporating more state-of-the-art baselines, such as LKH, would strengthen the comparison.

**Questions:**

1. Why not explore other TSP variants, such as dynamic TSP or dynamic CVRP?
2. Table 1 presents an equal number of static and dynamic vertices. How does performance vary with different numbers of static and dynamic vertices?

---

### Official Review · Reviewer_NPQy · 2024-11-04

**Soundness:** 3
**Presentation:** 3
**Contribution:** 2
**Rating:** 3
**Confidence:** 4

**Summary:**

The authors propose a neural combinatorial optimization method tailored to a newly introduced problem, the Partially Dynamic Traveling Salesman Problem (PDTSP). This problem resembles the traditional TSP but includes an added complexity: a subset of nodes is only revealed incrementally after some nodes have already been visited. The model architecture used is largely based on existing neural combinatorial optimization frameworks, with modifications to handle the new dynamic nature of node availability. Experimental results demonstrate that the proposed approach performs well on test instances, even surpassing a baseline that recalculates routes using Concorde each time new nodes are revealed.

**Strengths:**

- **Interesting Topic**:
The application of machine learning techniques to stochastic combinatorial optimization problems is still relatively underexplored. By addressing the PDTSP, this paper ventures into a promising direction, demonstrating the potential of ML in handling stochastic optimization problems.
- **Performance of Approach**:
The proposed approach achieves impressive results on the evaluated test sets, and it is noteworthy that the method can outperform the Concorde solver with recalculations after each decision step. This highlights the method's practical feasibility for handling dynamic elements in routing problems.

**Weaknesses:**

- **Limited Novelty**:
The methodological contribution of this paper is somewhat limited, as the model is primarily based on existing architectures with incremental modifications to accommodate the dynamic aspects of the PDTSP. The training process follows approaches similar to prior works, with only minor adjustments.
- **Insufficient Motivation for a New Problem Variant**:
Although the PDTSP is introduced as a new variant, the authors do not clearly explain why this specific variant is significant or necessary. Without a strong justification, it may appear that the problem was crafted to demonstrate ML strengths, rather than addressing a genuinely pressing real-world challenge. Additionally, the choice of abbreviation "PDTSP" is problematic, as this acronym is already associated with the Pickup and Delivery Traveling Salesman Problem. Using an already established abbreviation could cause confusion.
- **No Thorough Literature Review**:
The authors have not conducted a thorough literature review, which is a significant gap, especially given the existence of ML-based methods for handling stochastic optimization problems.
- **Focus on a Single Optimization Problem**:
The paper concentrates solely on the PDTSP without investigating whether the proposed method generalizes to more complex or varied routing problems. Given the range of challenges in real-world routing—often involving constraints like time windows, resource capacities, or multi-depot setups—it remains unclear whether this approach can extend to these scenarios.
- **No Ablation Studies**:
The paper lacks ablation experiments, which would be valuable for understanding the specific contributions of different components or modifications in the model architecture.

**Questions:**

.

---

### Meta-Review · Area_Chair_jTno · 2024-12-19

**Metareview:**

This paper proposed a learning based method to solve the partially dynamic TSP. In particular, some of the locations are known before solving, while some nodes dynamically arrive during the tour. Authors model the problem as a dynamic graph, and design the neural architecture based on existing Graph Convolutional Network based method. Reviewers agree that the strength of the paper is that it studies the (partially) dynamic version of TSP, which is of practical interest. However, they also raised serious concerns regarding the motivation, technical novelty, and evaluation of this paper. In particular, the motivation and practical setting of the partially dynamic TSP is insufficient; the proposed neural architecture is largely based on the GCN in [Joshi2022], and the evaluation should include more advanced baselines. Given the consistent negative evaluation from all reviewers, I recommend rejection.

**Additional Comments On Reviewer Discussion:**

While authors acknowledged the value of reviewers' comments, they did not provide meaningful response. So reviewers' opinions remain the same, which are all negative.

---

### Decision · Program_Chairs · 2025-01-22

Reject